# *DonnaRosa* Project: Exploring Informal Communication Practices Among Breast Cancer Specialists

**DOI:** 10.3390/curroncol32120704

**Published:** 2025-12-14

**Authors:** Antonella Ferro, Flavia Atzori, Catia Angiolini, Michela Bortolin, Laura Cortesi, Alessandra Fabi, Elena Fiorio, Ornella Garrone, Lorenzo Gianni, Monica Giordano, Laura Merlini, Marta Mion, Luca Moscetti, Donata Sartori, Maria Giuseppa Sarobba, Simon Spazzapan, Roberto Lusardi, Enrico Maria Piras

**Affiliations:** 1Medical Oncology and Rete Clinica Senologica, Azienda Provinciale per i Servizi Sanitari, 38122 Trento, Italy; 2Dipartimento di Scienze Economiche e Sociali (DiSES), Università Politecnica delle Marche, 60121 Ancona, Italy; f.atzori@staff.univpm.it; 3Breast Medical Oncology, Azienda Ospedaliero Universitaria Careggi, 50134 Firenze, Italy; angiolinic@aou-careggi.toscana.it; 4Medical Oncology Unit Ospedale di Montebelluna, ULSS 2 Marca Trevigiana, 31044 Treviso, Italy; michela.bortolin@aulss2.veneto.it; 5Medical Oncology Unit, Azienda USL-IRCCS di Reggio Emilia, 42122 Reggio Emilia, Italy; hbc@unimore.it; 6Precision Medicine Unit in Senology Unit, Fondazione Policlinico Universitario A. Gemelli IRCCS, 00136 Roma, Italy; alessandra.fabi@policlinicogemelli.it; 7Medical Oncology Unit, Azienda Ospedaliera Universitaria Integrata Verona, 37124 Verona, Italy; elena.fiorio@aovr.veneto.it; 8Medical Oncology Unit, Fondazione IRCCS Ca’ Granda Ospedale Maggiore Policlinico Milano, 20122 Milano, Italy; ornella.garrone@policlinico.mi.it; 9Struttura Semplice di Unità Operativa “Oncologia DH Cattolica”, 47841 Cattolica, Italy; lorenzo.gianni@auslromagna.it; 10Medical Oncology Unit, Sant’Anna Como, 47841 Como, Italy; monica.giordano@asst-lariana.it; 11Medical Oncology Padova Sud, 35131 Padova, Italy; laura.merlini@aulss6.veneto.it; 12Medical Oncology Ospedali di Cittadella e Camposampiero AULSS6 Regione Veneto, 35012 Camposampiero, Italy; marta.mion@aulss6.veneto.it; 13Oncology and Hemathology Department, Azienda Ospedaliero-Universitaria Policlinico di Modena, 41125 Modena, Italy; moscetti.luca@aou.mo.it; 14Medical Oncology Mirano-Dolo, 30035 Mirano, Italy; donata.sartori@aulss3.veneto.it; 15Medical Oncology ASL3 Nuoro, 08100 Nuoro, Italy; mariagiuseppa.sarobba@aslnuoro.it; 16Oncologia Medica e Prevenzione Oncologica Centro di Riferimento Oncologico (CRO), IRCCS Aviano, 33081 Aviano, Italy; spazzapan@cro.it; 17Dipartimento di Scienze Aziendali, Università degli Studi di Bergamo, 24129 Bergamo, Italy; roberto.lusardi@unibg.it; 18Digital Health & Wellbeing Center, Fondazione Bruno Kessler, 38122 Trento, Italy; piras@fbk.eu

**Keywords:** community of practice, oncologist, breast cancer, instant messaging apps, second opinion

## Abstract

In modern healthcare, clinicians increasingly use instant messaging apps such as WhatsApp^®^ to exchange information when institutional systems are slow or fragmented. *DonnaRosa* is an Italian community of breast cancer specialists who have used a WhatsApp^®^ group since 2017 to discuss complex cases, share clinical opinions, and provide mutual support. This study explores how informal digital communication can develop into a professional network that complements formal infrastructures. Analysis of survey responses and group interactions shows that the community facilitates quick decision-making, continuous learning, and reassurance, especially in smaller oncology units with limited expertise. Although participant selection and gender imbalance represent potential biases, the *DonnaRosa* experience highlights how grassroots collaboration can strengthen clinical practice, foster professional growth, and promote resilience. These findings emphasise the enduring importance of human networks in the evolving landscape of digital communication in oncology. However, the lack of social guidelines on the professional use of instant messaging apps raises important governance and accountability concerns, emphasising the need for structured recommendations.

## 1. Introduction

Communication technologies are increasingly central to healthcare, particularly after COVID-19 pandemics [1], shaping both patient–provider interactions, professional collaboration and even medical education [2,3]. In addition to official, institutionally supported information systems, general-purpose tools such as instant messaging applications (IMAs) are often used to complement data sharing, organisational coordination, and shared decision making. The spread of mobile phones, SMS, email, and IMAs has fostered what have been described as “minimal infrastructures” for healthcare communication, often bypassing traditional organisational systems [4]. Several studies have previously documented the use of IMAs for remote monitoring, professional networking, and clinical coordination, indicating meaningful impacts on workflow and referral patterns [5,6,7,8,9]. Such tools raise concerns about workload, reimbursement schemes, and data security. Concerns around record-keeping, confidentiality, and medico-legal responsibility have been repeatedly documented when clinicians use non-institutional IMAs [10]. However, such information-sharing practices reveal gaps in institutional infrastructures and point to strong capacity for bottom-up self-organisation, immediacy, collegiality, and resilience. We refer here to such technologically sustained arrangements as “grassroots infrastructures” [11], where the definitions of infrastructure, borrowed from the tradition of Science and Technology Studies, does not refer to the “pipes and wires” only but rather consider the interconnected network or assemblage of people, practices, artefacts, and institutions. We analyse this case as an example of a grassroots infrastructure and, crucially, as a Community of Practice (CoP). CoPs are professional networks created to sustain shared expertise, mutual learning, and critical dialogue [10,12,13]—forms of human-centred collaboration that stand in contrast to the logic of automation. While the use of IMAs among healthcare professionals is increasingly documented, little is known about how bottom-up CoPs operate within specific national oncology systems. Breast cancer care in Italy is characterised by a highly heterogeneous landscape of small- and medium-sized units, rapid evolution of complex clinical guidelines, and limited institutional mechanisms for obtaining real-time second opinions. These structural constraints contribute to professional isolation and potential variability in clinical interpretation. In this context, DonnaRosa emerges not merely as another example of IMA use, but as a grassroots CoP that functions as an adaptive response to systemic communication gaps. This study provides the first mixed-methods analysis of such a community within Italian breast oncology, addressing a gap in both CoP and IMA literature.

*DonnaRosa (Pink Lady, in Italian)* is a WhatsApp^®^-based community of breast cancer specialists founded in 2017 by seven oncologists.

The group was originally established to facilitate the exchange of expert opinions on anonymised patient cases in real time. However, it has since evolved into a platform for daily peer-to-peer consultations and it grew up to create a network for oncologists who confer with one another in complex clinical cases. The community has expanded to comprise 54 members working in Italy and one in France, the growth being facilitated by an informal, invitation-based recruitment process. The community now functions as a forum for the dissemination of research findings, administrative updates and clinical trial opportunities. The members convene annually for an in-person workshop, hosted in a different Italian location each time. However, the focal point of the community is the WhatsApp^®^ group chat. Due to the characteristics of its recruitment, and limitations of the platform, there is an absence of centralised oversight or formal record of participation or activity. Since its inception, the core of the community is a group chat on WhatsApp^®^. The group chat was initially used solely to share anonymised patients’ data to have an expert peer opinion in real time. Over time, the group chat became used to share research papers, administrative information, clinical trial enrolling requests. Because of the open recruitment model—where any member can freely invite others—and the use of WhatsApp^®^ as the hosting platform, there is no central oversight or systematic record of either membership or user activity within the group. This study represents our first systematic attempt to map the community through a survey, in order to better understand the nature and identity of the community its and the ways in which members interact with the grassroots infrastructure

## 2. Materials and Methods

This analysis constitutes a component of an ongoing interdisciplinary research project, which involves medical researchers (oncologists) and social scientists (sociologists). The research has been structured in three phases. The first phase consisted of a preliminary exploration of the information-sharing practices of the community, carried out through a textual analysis of the group’s messages [11] and a focus group with the founding members. Findings from the initial phase guided the creation of an “ad hoc” questionnaire on WhatsApp^®^, use, which is the central focus of this paper, as well as the design of a series of in-depth interviews, which are currently underway. The objective of these interviews is to further explore the community history, its evolution over time, and how members perceive and make sense of this grassroots infrastructure.

This study adopts a mixed-methods approach, combining qualitative and quantitative data collection and analysis:

### 2.1. Document Analysis

The log files of messages exchanged via the IMA were analysed to identify interaction patterns, language use, and implicit rules of engagement within the group. This initial analysis was undertaken after the inaugural *DonnaRosa* meeting, held in November 2019.

### 2.2. Survey

In August 2024 a structured online questionnaire was distributed to all members of the WhatsApp^®^ group, which at the time consisted of 54 members who had been invited or recruited through existing professional or personal networks, with the aim of ensuring diversity in workplace settings (Scientific Institute for Research, Hospitalization and Healthcare (IRCCS), general hospitals, and other institutions), age, and clinical experience —factors that may influence how clinical information is interpreted. In addition, concerted efforts were made to include clinicians with varying levels of expertise, thereby reflecting the widespread distribution of oncology services across the country. In such a context, professionals are often required to address a broad spectrum of patient needs. It was considered prudent to exclude opinion leaders and individuals whose authority might inhibit open discussion. The survey explored demographic information, recruitment processes, usage habits, and perceptions of the utility of the *DonnaRosa* community. Because membership was based on personal invitations and professional relationships, the sample may be subject to selection bias. To mitigate this risk, survey results were triangulated with objective chat activity metrics, and comparisons were made between high- and low-frequency participants. We also acknowledge that affinity-based networks may inflate perceptions of usefulness, and we interpret survey findings with this consideration in mind.

### 2.3. Planned Interviews (Ongoing)

A series of semi-structured interviews with founding members and selected participants is currently underway. The objective of these interviews is to explore the community’s origins, recruitment strategies, norms of interaction, and its perceived role in clinical and professional practice.

## 3. Results

### 3.1. Qualitative Findings

The WhatsApp^®^ group was conceived and initiated by two breast cancer specialists (AF and DS) with the aim of facilitating the exchange of perspectives on controversial clinical cases. Within a few days, five additional oncologists joined the initiative, embracing its vision and objectives. In the following years, the founding members involved another 47 colleagues, gradually expanding the community. There was no predefined minimum or maximum number of participants. All members admitted to the group were encouraged by the colleague who had introduced them to actively participate in the community’s activities and exchanges. The recruitment process was performed in a consecutive manner and was entirely independent of any personal involvement. Nonetheless, the objective was to maintain a relatively limited group size, with the aim of circumventing management challenges and facilitating meaningful interaction. The growth of the *DonnaRosa* community has occurred through personal invitations, leading to a diverse group of many specialists who are not necessarily personally acquainted to all the group members. This informal recruitment model, combined with the non-binding and accessible nature of WhatsApp^®^, fosters flexible and self-regulated participation.

The following conclusions were drawn from the analysis:The interaction style is characterised by the utilisation of concise, colloquial messages frequently without punctuation. The patient’s identity is concealed and data shared do not allow to identify them. Emoticons and an emphatic tone are also employed with regularity. As illustrated in Figure 1, the screenshots depict a selection of conversations from the quick chat function.

2.The following patterns of use have been identified: firstly, clinical case discussions and requests for reassurance, and secondly, exchanges about guidelines, trial recruitment, and administrative issues.3.In order to prevent any disruption to proceedings, founders intervened to discourage any off-topic or inappropriate contributions.4.In order to establish a written record of the messages, the utilisation of voice messages was intentionally avoided5.Second opinions ranged from simple confirmations to extended deliberations that reflected a diversity of local practices.6.The group facilitated the exchange of trial opportunities, best practices and evolving clinical protocols, thereby enabling members to expand their knowledge and expertise beyond the scope of second opinions.

### 3.2. Survey Findings

The survey, which was completed by 92.5% (50 out of 54) of participants, provides the first systematic overview of the *DonnaRosa* community. The *DonnaRosa* group is predominantly composed of participants over the age of 45, the 80% of them having 11+ years of experience in the field. Table 1 reports demographic characteristics of the members of the chat group. Members engaged with the chat at varying levels, based on individual needs and availability. We observed that many participants engage with the chat primarily when clinical decisions are being discussed, highlighting its importance as a collaborative tool in complex or uncertain situations. A salient feature of this phenomenon is the absence of any correlation between participation and professional hierarchy. Irrespective of whether an individual holds the title of Director of a Complex Unit or occupies a different role, there is a pervasive sense of collective contribution, with many individuals contributing based on their competence rather than their job title. In fact, most contributors stated that they give their opinion only if they feel competent on the topic, and sometimes only if the case has not already been widely discussed. Others participate when they believe they can add something meaningful to what has already been said. Some prefer to step back and allow those with more expertise to speak first, which shows a clear respect for knowledge and experience within the group. In terms of frequency, participation varies—ranging from a few times a month to once or several times a week, depending on the individual. But what remains constant is that *DonnaRosa* is used as a space where competence is valued more than hierarchy (see Table 2).

Survey responses indicate that the primary perceived value of the chat (see Figure 2) lies in sharing of second opinions, particularly for complex or ambiguous clinical cases. Additionally, several participants underscored the importance of the platform in facilitating the exchange of best practices and enabling collaboration in clinical trials. The chat also serves, albeit to a lesser extent, as a space for discussing administrative and organisational issues. Members frequently request guidance when they perceive the guidelines to be lacking in comprehensiveness (54%), insufficiently updated (44%), or ambiguous (58%) guidelines. In more than half of the cases, individuals request counsel on ethical dilemmas associated with the specific circumstances. Many participants reported that receiving suggestions from the group often led them to reconsider their initial opinions, provided reassurance, or allowed them to view the clinical issue from a new perspective. Regarding clinical decision-making, most participants described a common sequence of consultation: first, reviewing guidelines and scientific evidence in the literature; second, consulting colleagues within their own department (when feasible); third, turning to the *DonnaRosa* chat; and finally, seeking a second opinion from an external expert. As reported in Table 3, many participants (54%) tend to trust collective agreement, likely because it provides a sense of shared validation and reduces individual decision-making risk. Furthermore, 40% of respondents indicated that they place greater importance on the experience and professional expertise of the colleague providing the advice. This finding is of particular significance, as it underscores the notion that perceived competence exerts a substantial influence on decision-making, nearly equal to that of the prevailing consensus.

## 4. Discussion

Several studies have examined the use of instant-messaging applications (IMAs) in healthcare, yet few provide systematic analyses of oncology-focused communities of practice that have developed organically over time. The narrative review by Gebbia [14] and colleagues demonstrates that WhatsApp is widely used by oncology professionals as a “quick and easy communication tool,” but also underscores major concerns regarding data security, privacy, medico-legal responsibility, and the absence of standardised guidelines for clinical use. Additional reports—including WhatsApp-based observational analyses during the COVID-19 emergency in Italy [15], surveys of interprofessional communication among Lebanese physicians [16], and investigations of behavioural determinants of WhatsApp use within Italian hospitals (De Benedictis et al.) [5]—confirm that IMAs are pervasive and valued for improving workflow efficiency and access to clinical expertise, while simultaneously raising ethical, organisational, and regulatory questions. Systematic reviews of broader social-media use among healthcare professionals [17] similarly highlight benefits for networking, professional collaboration, and education, but emphasise persistent concerns regarding privacy, liability, and data protection. To our knowledge, no published studies describe an oncology-specific CoP in the Italian context with characteristics directly comparable to *DonnaRosa* in terms of size, longevity, and sustained clinical activity.

Notably, to the best of our knowledge, no clinical guideline issued by major oncology societies—including NCCN, ESMO, ASCO, or AIOM—has ever addressed the use of IMAs for communication among healthcare professionals, despite their increasing ubiquity in daily practice. Existing guidelines from scientific societies have traditionally focused on physician–patient communication (e.g., diagnostic disclosure, prognosis discussion, therapeutic decision-making, end-of-life communication, and training in communication skills), as exemplified by the ASCO Consensus Guideline on Patient–Clinician Communication [18].

Given these risks and the increasing clinical relevance of informal messaging networks, a further consideration concerns the potential role of national and international professional societies in establishing governance frameworks for their safe and appropriate use. As informal digital communication networks increasingly shape day-to-day clinical reasoning and real-time decision-making, the need for structured oversight becomes evident. In this context, national and international oncology societies are well positioned to provide authoritative guidance on the safe and appropriate use of instant-messaging applications (IMAs) within professional settings. Recent evidence, including the scoping review by Morris [9], highlights critical challenges related to record keeping, data storage, and the absence of mechanisms for traceability and clinical documentation when WhatsApp is used for professional communication. Such concerns reinforce the urgency of establishing minimum standards for privacy and cybersecurity, clear expectations regarding the documentation of clinical recommendations arising from informal exchanges, and explicit boundaries between informal peer consultation and formal clinical advice. Furthermore, professional bodies could support the evaluation and adoption of secure enterprise-grade communication platforms as alternatives to consumer applications, thereby reducing medico-legal vulnerabilities. The establishment of coherent governance frameworks would allow clinicians to retain the agility and collaborative advantages of organically developed networks while mitigating the recognised risks related to data protection, liability, and ethical practice.

While the use of IMAs in clinical settings has attracted some interest, little attention has been paid to the use of such technologies for community-building purposes [19]. The case presented here shows how IMAs can foster peer participation and knowledge creationg and sharing. CoPs have a pivotal function in the generation and dissemination of knowledge, frequently tacit, whilst concurrently attenuating the risk of professional de-skilling and cultivating empathy among practitioners. Within these communities, the sharing of expert opinions can be understood as an infrastructuring practice. This is defined as an ongoing, collaborative process through which clinicians co-construct the communicative and informational infrastructure necessary for their work. As previously described, infrastructuring emphasises how users collaboratively design, adapt, and sustain information infrastructures over time [13]. In this context, the case of *DonnaRosa* illustrates how oncology specialists engage in a form of infrastructuring through everyday interactions, including the asking of questions, the provision of reassurance, the sharing of clinical reasoning, and the negotiation of professional boundaries. This collaborative dynamic exemplifies how expertise is not merely applied through adherence to protocols, but is continuously reinterpreted in relation to individual patient needs and local clinical realities. The *DonnaRosa* group demonstrates the potential for clinicians to repurpose general-purpose IMA into meaningful infrastructures of care. Although not formally recognised by institutional bodies, these grassroots infrastructures complement formal systems by offering immediacy, collegial reassurance, and professional solidarity [20].

When viewed through the lens of a CoPs, *DonnaRosa* emerges as a human-centred alternative to more technocratic or automated models of care. The programme fosters a conducive environment in which clinicians at all levels feel empowered to pose questions, share insights, and assist one another, ultimately contributing to enhanced patient care. The majority of members regard the platform as an effective instrument for obtaining second opinions and benchmarking decisions, thereby underscoring a shared sense of purpose grounded in common scientific principles and professional motivation. In addition to clinical support, the group has expanded its scope by fostering collaboration on research initiatives, paper writing, and conference presentations. This evolution serves to reinforce its identity as both a professional and social network committed to collective growth and scientific advancement. More broadly, the group facilitates key processes that are central to resilient healthcare practice. The provision of support from colleagues and the undertaking of collective reflection have been demonstrated to assist in the reduction in uncertainty in cases of a complex nature. Furthermore, these practices have been associated with the mitigation of professional isolation and the reduction in the risk of burnout [12,21,22]. Furthermore, knowledge circulates dynamically within the group, with members exchanging tacit expertise, clinical guidelines, and research opportunities. This contributes to a distributed and evolving knowledge base [23].

A further dimension emerging from our analysis concerns the internal organisational dynamics of the group. While the *DonnaRosa* community was intentionally designed as a non-hierarchical, peer-based space, our findings indicate that hierarchy is not absent but rather transformed. Specifically, we observed a clear distinction between formal hierarchy—rooted in institutional positions such as departmental leadership—and an informal, competence-based hierarchy grounded in peer-recognised expertise. Although no direct correlation was found between formal seniority and frequency of participation, members consistently reported deferring to colleagues perceived as more experienced or knowledgeable. This pattern suggests the emergence of an experience-driven leadership, shaped not by official roles but by accumulated clinical expertise, reliability in judgement, and the ability to articulate reasoning in a way that resonates with peers. Such emergent leadership influences conversational norms, guides the interpretation of complex cases, and supports the community’s cohesion. Rather than functioning as a flat network, *DonnaRosa* exemplifies a form of relational professionalism in which authority is negotiated dynamically and anchored in competence, illustrating how informal clinical networks can develop their own internal epistemic hierarchies.

Although participation did not correlate with formal seniority, our findings show that competence-based hierarchy plays a significant role in shaping interactions. Many members express a preference for contributions from more experienced colleagues, and high-competence individuals—who are often also senior clinicians—appear to exert significant influence. Thus, while the grassroots nature of the group reduces organisational authority, it does not eliminate hierarchical dynamics; instead, influence becomes grounded in peer-perceived expertise rather than institutional rank

*DonnaRosa* group adopts a human-centred approach that contrasts with automation by highlighting the value of critical judgement, empathy, and context-aware reasoning This approach serves to counteract prevailing trends towards algorithmic dependence and the erosion of clinical autonomy. The findings of this study serve to reinforce the significance of CoPs in oncology, highlighting their role as essential spaces for ongoing professional development. The role of these committees is to ensure the maintenance of clinical expertise, to encourage reflective practice, and to collectively uphold human agency in medical decision-making. When adequately supported, CoPs have been shown to enhance knowledge transfer, professional growth, and quality of care across healthcare settings [12]. Simultaneously, the case of *DonnaRosa* emphasises the indispensable nature of professional expertise. The strength of this group lies in the fact that its members are experienced clinicians who can draw on many years of professional knowledge and clinical expertise to inform their approach to guideline interpretation. Their clinical judgement and understanding of the contexts to which the guidelines apply is highly developed, contributing to effective and appropriate patient management. This human expertise remains the cornerstone of safe and meaningful care.

In the current technological landscape, the increasing prevalence of artificial intelligence (AI)—particularly large language models (LLMs)—presents both challenges and opportunities for professional communities [24]. While AI should not replace clinical expertise, it offers powerful tools to accelerate access to scientific literature and to retrieve complex data, such as subgroup analyses from clinical trials, that would otherwise be time-consuming to gather manually [25,26]. In this context, AI may serve as an auxiliary instrument that supports the identification, synthesis, and contextualisation of information, while retaining responsibility for interpretation and application within the professional community [27,28,29]. From this perspective, CoPs such as *DonnaRosa* may become even more relevant in the AI era. They provide a human-centred environment in which algorithmically retrieved data can be critically evaluated, confronted with real-world clinical experience, and balanced against patient-specific considerations. Rather than competing with professional expertise, AI should be approached as an auxiliary instrument—a tool that enhances the capacity of specialists to reason, compare, and decide together.

The synergy that emerges from the convergence of professional competence and computational capacity may signify a productive path forward. The utilisation of AI facilitates the acquisition of information with both expediency and breadth, while CoPs ensure the preservation of depth, nuance, and the safeguarding of human judgement.

Such a dual infrastructure—technological and human—could be particularly valuable in complex fields such as oncology, where treatment choices are rarely straightforward and where the combination of evidence-based knowledge and professional dialogue remains crucial. This could also serve as a premise and a strategy to be adopted within *DonnaRosa* in the coming years, aiming to make the CoP more aware and better prepared to face the rapid evolution of decision-making algorithms in oncology, encompassing diagnostic, therapeutic, and rehabilitative aspects. This reflection on a ‘dual infrastructure’—human and technological—should be interpreted as a forward-looking proposition rather than as a conclusion derived from empirical findings. The present study did not assess the use or perception of AI/LLM tools within the community. Future work will address this gap by surveying members about current and potential uses of AI, piloting the integration of AI-assisted evidence summarisation within the CoP, and studying how such tools might complement or reshape peer-supported decision-making processes.

### Limitations and Interpretative Considerations

The present study is subject to several limitations. Firstly, the development of the *DonnaRosa* community was undertaken without the involvement of international key opinion leaders. This was a deliberate choice, aimed at promoting discussion among Italian experts, whose clinical decision-making is shaped by national guidelines and context-specific regulatory requirements. While this approach fostered a collegial and non-hierarchical environment, it may also have limited the range of perspectives represented and introduced selection biases, including potential geographic imbalances. Furthermore, recruitment was primarily based on pre-existing professional or personal relationships, which may have reinforced implicit affinities or preferential networks rather than ensuring systematic representativeness. Another limitation pertains to the demographic composition of the group. The marked female predominance (82%) may partly reflect the current feminisation of oncology in Italy, but it could also indicate a gender-related selection bias. Nonetheless, this feature may carry positive implications. Previous evidence suggests that female physicians may exhibit greater empathy and patient-centred communication, factors which are associated with improved outcomes [30]. Furthermore, the mean age of participants (51.4 years) was found to be marginally higher than that expected for frequent users of digital tools. This discrepancy is presumably attributable to the initial recruitment choices, which placed a premium on experienced clinicians. Paradoxically, this reinforces the notion that digital communication in healthcare is not limited to younger generations but may also serve as a channel for senior professionals seeking rapid, collegial exchange. Among the strengths of the initiative, the diversity in workplace settings (IRCCS, general hospitals, and other institutions), age, and clinical experience enriched discussions by bringing multiple perspectives to the interpretation of clinical data. The chat proved to be of particular value in the Italian oncological context, where clinicians often manage multiple tumour types within small units, sometimes without local colleagues specialised in a given pathology. In such cases, the group provided immediate access to external expertise—a form of “distributed consultation” that compensated for institutional asymmetries and strengthened professional support across the country.

Qualitative insights derived from chat logs primarily reflect the behaviours of active contributors, who represent approximately one-third of the group. Interaction patterns may therefore not capture the perspectives of low-frequency or silent members, and are best interpreted as illustrative of dominant norms rather than universally shared practices.

## 5. Conclusions

*DonnaRosa* provides an illustrative example of how IMAs, when adopted by healthcare professionals, can evolve into robust grassroots infrastructures and effective CoPs. This case exemplifies the creativity and pragmatism of clinicians in addressing gaps in institutional communication, while also offering a model of resistance to over-reliance on automation. The provision of support for such initiatives has the potential to inform the design of future healthcare communication systems. Indeed, such support could combine the flexibility of grassroots practices with the guarantees of institutional infrastructures. Ultimately, the findings of this study reaffirm the importance of CoPs as vital spaces for continuous professional development, the preservation of clinical expertise, and the collective safeguarding of human judgement in medical decision-making. Systematic reviews consistently highlight that when appropriately supported, CoPs improve knowledge translation, enhance professional learning, and contribute to better clinical practice [8].

In the future, the integration of AI—in particular, LLMs—presents opportunities to expand access to literature and trial data, provided that it is critically evaluated within human networks. In this context, AI should be regarded as an auxiliary tool that complements, rather than replaces, clinical judgement. For *DonnaRosa*, adopting this perspective could represent a strategic path in the coming years, enabling the CoP to remain agile and prepared for the rapid evolution of diagnostic, therapeutic, and rehabilitative decision-making algorithms, while safeguarding the central role of professional expertise.

The experience of the *DonnaRosa* community highlights both the value and the vulnerabilities of organically developed digital networks, reinforcing the need for structured guidance from professional societies to ensure safe, ethical, and sustainable integration into oncology practice.

## Figures and Tables

**Figure 1 curroncol-32-00704-f001:**
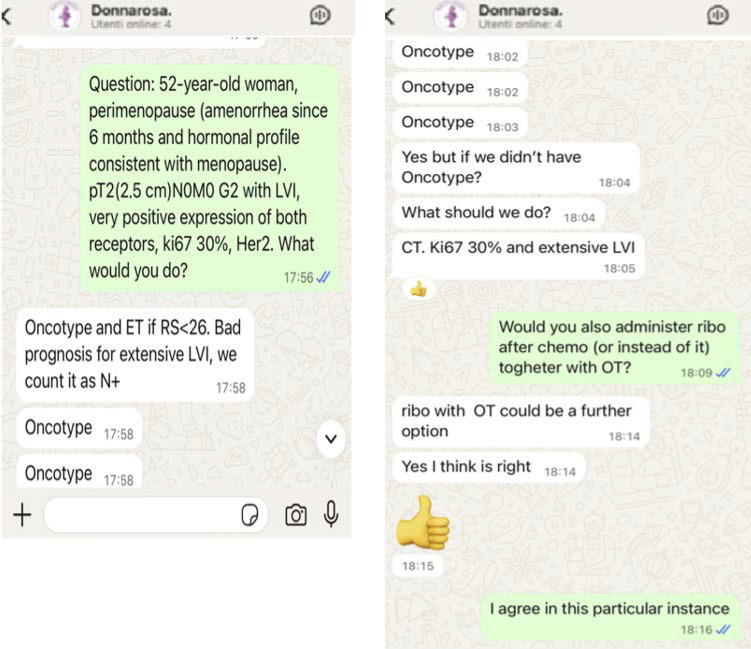
Selection of some screenshots from the *DonnaRosa* chat about a clinical case. Key to abbreviations used: N = node; G = grading; LVI = Lymphovascular Invasion; ET = Endocrine Therapy; RS = Recurrence Score; Ki67 = Proliferation marker protein Ki-67; HER2 = Human Epidermal growth factor Receptor 2; Chemo = Chemotherapy.

**Figure 2 curroncol-32-00704-f002:**
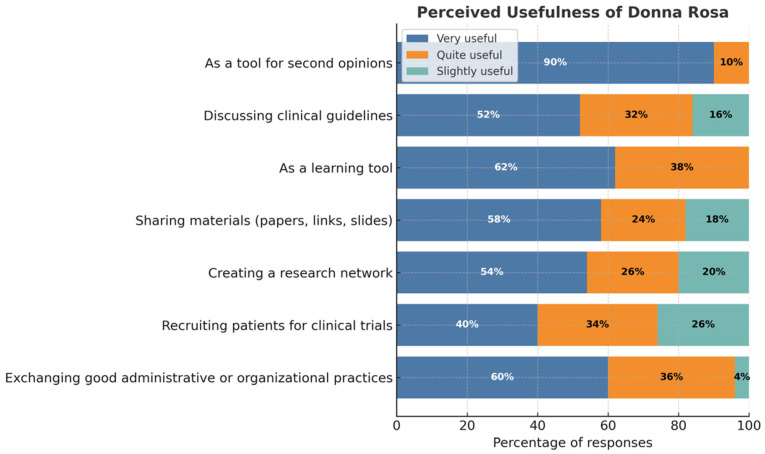
Perceived value of the chat.

**Table 1 curroncol-32-00704-t001:** Demographic characteristics of the *DonnaRosa* members.

Characteristics	Respondents N 50 (100%)
Sex	
Female	41 (82%)
Male	9 (18%)
Age	
31–40	7 (14%)
41–50	17 (34%)
51–60	16 (32%)
61–70	10 (20%)
Mean age (y)	51.44
Years of service	
1–5	7 (14%)
6–10	3 (6%)
11–20	16 (32%)
21–30	19 (38%)
>30	5 (10%)
Professional role	
Head of complex unit	10 (20%)
Head of simple unit	10 (20%)
Medical officer	28 (56%)
Researcher	2 (4%)
Number of oncologists in the affiliated institution	
1–5	21 (42%)
6–10	11 (22%)
11–15	5 (10%)
15–20	4 (8%)
>20	9 (18%)

**Table 2 curroncol-32-00704-t002:** Patterns of participation in the *DonnaRosa* community.

Variable	Response Options	% of Participants
Main mode of participation	Reading messages mainly when clinical decisions are discussed	78%
	Asking for advice on a clinical case	31%
	Offering advice on a clinical case	36%
Motivation to contribute	Give opinion only if competent	64%
	Give opinion if competent and case not already discussed	36%
	Prefer that more experienced colleagues speak first	22%
	Intervene only if can add something meaningful	42%
Frequency of participation	Never/<once a month	28%
	Once or a few times per month	42%
	Once or several times per week	30%
Professional role	Director of Complex Unit	22%
	Director of Simple Unit	19%
	Medical Executive	53%
	Researcher	6%
Key feature	Correlation between hierarchy and participation	None observed

**Table 3 curroncol-32-00704-t003:** Primary criterion considered when receiving peer suggestions.

	N	%
The opinion expressed by the majority of respondents	27	54%
The experience and professional expertise of the colleague providing the advice	20	40%
Direct personal knowledge of the colleague providing the advice	2	4%
Both majority opinion and individual expertise	1	2%
Total	50	100%

## Data Availability

The original contributions presented in this study are included in the article. Further inquiries can be directed to the corresponding author.

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
