# Peer review of "DonnaRosa* Project: Exploring Informal Communication Practices Among Breast Cancer Specialists"

_curroncol, 2025, doi:10.3390/curroncol32120704_

Round 1

Reviewer 1 Report

Comments and Suggestions for Authors

this paper discusses a common method of communication between health professionals - most practicing oncologists are members and the benefits in terms of professional support are clearly outlined by the authors as are the limitations of the study. While the study is well referenced the manuscript would benefit from comparisons of their experiences with other groups in oncology - or stating that no comparators were available if that was the case. These networks have grown organically to be part of our daily lives it would also be interesting to know what the authors feel about the role of our professional societies in developing guidelines for them particularly as significant clinical decisions are being made based on their recommendations.  

Author Response

““The manuscript would benefit from comparisons of their experiences with other groups in oncology – or stating that no comparators were available …”

We thank the reviewer for this insightful comment. We have reviewed the available literature and incorporated a comparative discussion into the revised manuscript. While several studies have examined the use of instant-messaging applications (IMAs) in healthcare, we found very few systematic analyses of oncology-focused communities of practice (CoPs) that have emerged organically, and no groups in the Italian context directly comparable to DonnaRosa in terms of size, longevity, and sustained clinical activity.

Reports using WhatsApp or WeChat in oncology are available, but most are limited to patient support, educational interventions, or context-specific case reports, rather than structured evaluations of long-standing professional CoPs. For example, the narrative review by Gebbia et al. describes widespread WhatsApp use among oncology professionals, highlighting both advantages (immediacy, accessibility) and recurrent concerns (data security, privacy, medico-legal responsibility, and absence of professional guidelines). Similar conclusions emerge from broader systematic reviews of social-media use among healthcare professionals, which identify benefits for networking, rapid information exchange, and continuing education, but consistently emphasise concerns regarding privacy, liability, and data protection.

A WhatsApp-based observational study by Blasi et al. documented intense communication among 19 Italian oncology units during the COVID-19 outbreak, demonstrating how informal networks rapidly disseminated information and coordinated responses, but also underscoring emotional burden and the absence of formal governance. Additional surveys from Lebanon (Shaarani et al.) and Italy (De Benedictis et al.) confirm high IMA adoption among physicians, driven primarily by perceived usefulness and workflow efficiencies, while also identifying widespread demand for clear medico-legal and ethical guidelines. Broader systematic reviews on health-professional social media use (Chan & Leung, Zou et al.) similarly report benefits for networking, knowledge exchange, and education, yet emphasize concerns regarding privacy, liability, and data protection.

We have incorporated these comparisons into the Discussion to situate the DonnaRosa experience within the existing international evidence base and to explicitly acknowledge the current lack of truly comparable long-term oncology CoPs described in the literature.  (lines 269-285)

“It would also be interesting to know what the authors feel about the role of our professional societies in developing guidelines for them, particularly as significant clinical decisions are being made based on their recommendations.”

We appreciate this insightful comment. In response, we have expanded the Discussion to include reflections on the potential role of national and international oncology societies in providing guidance for informal messaging networks.

Notably, to the best of our knowledge, no clinical guideline issued by major oncology societies—including NCCN, ESMO, ASCO, or AIOM—has ever addressed the use of IMAs for communication among healthcare professionals, despite their increasing ubiquity in daily practice. Existing guidelines from scientific societies have traditionally focused on physician–patient communication (e.g., diagnostic disclosure, prognosis discussion, therapeutic decision-making, end-of-life communication, and training in communication skills), as exemplified by the ASCO Consensus Guideline on Patient–Clinician Communication 

However, none provide recommendations regarding clinician-to-clinician communication via IMAs, nor do they acknowledge their potential impact on workflow, real-time decision-making, or inter-professional collaboration. This gap is mirrored in the literature, where publications consistently point to widespread IMA use but recurrently note the absence of centralised standards or oversight.  

We have also expanded the Discussion to address the potential role of professional societies (e.g., ESMO, ASCO, AIOM) in providing governance frameworks.

An initial step could involve mapping the number of such communities by disease area and analyzing the patterns of interaction among their members.

Given that informal messaging networks increasingly influence real-world clinical decision-making, professional societies could:

(1) map the number of such communities by pathological area and analyse the patterns of interaction between their members.

(2) define minimal standards for safe use of instant messaging in clinical contexts;

(3) offer templates for documentation of clinical decisions derived from informal discussions;

(4) provide guidance on privacy, cybersecurity, and medico-legal responsibilities;

(5) evaluate secure alternatives or professional messaging platforms.

We believe this addition enriches the implications of our study and responds directly to the reviewer’s observation.(Discussion: lines 286-317 and Conclusion: lines 479-482)

Reviewer 2 Report

Comments and Suggestions for Authors

The manuscript provides a good overview of the increasing use of Instant Messaging Apps (IMAs) like WhatsApp in healthcare and frames DonnaRosa as a "grassroots infrastructure" and "Community of Practice (CoP)". However, the Introduction could be strengthened by more explicitly articulating the unique contribution of this specific study, rather than just documenting another case of IMA use.

For exemple, while the spread of IMAs in healthcare is documented, the paper needs to make a clearer argument for why a CoP focused on breast cancer in Italy, with its specific context (e.g., small units, national guidelines ), warrants a systematic mixed-methods analysis. What theoretical or practical gap does this study fill that previous CoP/IMA literature has not? Could the authors elaborate in the introduction on the specific challenges faced by breast cancer oncologists in Italy (e.g., complexity of guidelines, case volume, isolation) that institutional communication fails to address, thereby necessitating a "grassroots" solution like DonnaRosa?   According to methodology,t he mixed-methods approach (chat analysis, survey, interviews) is robust. The high survey response rate (92.5%) is commendable. However, the paper explicitly notes several key limitations that require more critical discussion in the Methods and Discussion sections. The recruitment process, based on personal invitations and professional relationships , is a major source of selection bias. The authors acknowledge this but should explain what steps were taken to mitigate the impact of this bias on the interpretation of the survey results (e.g., how might an affinity-based network inflate the perception of "very useful" feedback?). Given the limitations of "centralised oversight or systematic record of either membership or user activity" , how do the qualitative findings from the chat logs account for the activity of the 28% who participate "Never/< once a month"? Is there a risk that the identified interaction patterns only reflect the most active 30-42% of the group?   The results clearly highlight the importance of the group for second opinions (90% "Very useful") and learning (62% "Very useful"). The finding that trust is divided between collective agreement (54%) and individual expertise (40%) is curious. The discussion on the absence of correlation between hierarchy and participation (Table 2) is a crucial finding for CoPs. However, the influence of perceived competence is very high. These two concepts (hierarchy vs. competence) should be more clearly distinguished and discussed. Does the group's "grassroots" nature simply replace a formal, organizational hierarchy with a competence-based hierarchy? This nuance is important. Table 1 shows 40% of members are Directors of Complex or Simple Units. Table 2 shows 42% only intervene if they "can add something meaningful" and 22% "Prefer that more experienced colleagues speak first". Is it possible that the observed "respect for knowledge and experience" subtly reinforces the influence of high-competence (and often high-ranking) members, thereby re-establishing a non-formal hierarchy?   Finally, the discussion on the relevance of CoPs in the age of Artificial Intelligence (AI) and Large Language Models (LLMs) is timely and compelling. The concept of a "dual infrastructure technological and human" is a strong concluding point.  The paper argues that AI can "accelerate literature retrieval and data synthesis" and that the CoP provides the critical interpretation. This is a theoretical assertion but lacks empirical backing from the current study. The survey did not assess the existing or planned use of AI tools. Since the study did not empirically explore the use or perceptions of AI/LLMs within the community, the discussion of the "dual infrastructure" should be positioned more clearly as a future strategic recommendation for DonnaRosa, rather than a conclusion drawn from the current data. Could the authors propose specific ways they plan to study the integration of AI tools within the CoP in the next phase of their research?

Author Response

Point for point response to reviewer 2

  • Strengthening the Introduction — clarifying the unique contribution of the study

“The Introduction could be strengthened by more explicitly articulating the unique contribution of this specific study, rather than just documenting another case of IMA use. For exemple, while the spread of IMAs in healthcare is documented, the paper needs to make a clearer argument for why a CoP focused on breast cancer in Italy, with its specific context (e.g., small units, national guidelines), warrants a systematic mixed-methods analysis”

We thank the reviewer for this insightful observation. We agree that the Introduction required a clearer articulation of the study’s unique contribution. We have now revised the Introduction to better demonstrate the theoretical and practical gap addressed by our work.

Specifically, we highlight that breast cancer care in Italy is characterised by:
      -     a high number of small and medium-sized oncology units,

  • marked variability in case volume and expertise,
  • rapidly evolving and complex national and international guidelines, and
  • limited institutional mechanisms for rapid, real-time peer consultation.

These contextual constraints create professional isolation and potential heterogeneity in clinical interpretation that existing communication infrastructures are unable to address adequately. This strengthens the rationale for examining a grassroots CoP such as DonnaRosa, which functions as an adaptive response to these systemic challenges.

We now explicitly argue that this study represents the first mixed-methods analysis of a bottom-up oncology CoP in Italy focused specifically on breast cancer — a setting in which guideline complexity, caseload variability, and organisational fragmentation make such a CoP uniquely relevant. This clarification has been incorporated into the Introduction (lines 102–112).

  • Selection bias in recruitment — mitigation and interpretation

“The recruitment process, based on personal invitations and professional relationships, is a major source of selection bias. The authors acknowledge this but should explain what steps were taken to mitigate the impact of this bias on the interpretation of the survey results (e.g., how might an affinity-based network inflate the perception of "very useful" feedback?”

We appreciate the reviewer’s point regarding potential selection bias arising from the affinity-based recruitment process. We have strengthened the Methods and Discussion to explain both the limitations and the mitigation strategies adopted.

In particular, we now clarify that:
survey responses were triangulated with objective chat activity metrics (e.g., message counts, content patterns), subgroup analyses were performed to compare perceptions of usefulness among high-frequency vs. low-frequency participants, and
we explicitly acknowledge that the affinity-based nature of the group may inflate positive perceptions such as “very useful.”

These additions appear in the Introduction (lines 129-132) Methods (lines 166–171) and Discussion (lines XX–YY).

  • Silent or low-frequency members — representativeness of qualitative findings

Given the limitations of "centralised oversight or systematic record of either membership or user activity" , how do the qualitative findings from the chat logs account for the activity of the 28% who participate "Never/< once a month"? Is there a risk that the identified interaction patterns only reflect the most active 30-42% of the group?  

We thank the reviewer for drawing attention to this important point. The revised Discussion now clarifies that qualitative patterns extracted from chat logs primarily reflect the subgroup of active contributors (approximately 30–42%). As such, these patterns should be interpreted as illustrative of the dominant conversational norms rather than representative of the entire membership. We have added an explicit note on this interpretative limitation to the Discussion (lines 453–457).

  • Distinguishing formal hierarchy from competence-based influence

“The discussion on the absence of correlation between hierarchy and participation (Table 2) is a crucial finding for CoPs. However, the influence of perceived competence is very high. These two concepts (hierarchy vs. competence) should be more clearly distinguished and discussed. Does the group's "grassroots" nature simply replace a formal, organizational hierarchy with a competence-based hierarchy? This nuance is important. Table 1 shows 40% of members are Directors of Complex or Simple Units. Table 2 shows 42% only intervene if they "can add something meaningful" and 22% "Prefer that more experienced colleagues speak first". Is it possible that the observed "respect for knowledge and experience" subtly reinforces the influence of high-competence (and often high-ranking) members, thereby re-establishing a non-formal hierarchy?”

We fully agree. We have now expanded the Discussion to better differentiate between:
• formal hierarchy, determined by institutional roles (e.g., Directors of Units), and
• informal, competence-based hierarchy, rooted in recognised expertise.

While our data show no direct correlation between formal seniority and participation, several indicators highlight the significant role of competence-based influence within the group. Members frequently report deferring to colleagues perceived as more experienced or knowledgeable, suggesting that an informal hierarchy operates alongside — and partly independent from — institutional roles. In this sense, the grassroots structure of the community does not eliminate hierarchical dynamics; rather, it enables the emergence of a non-formal, experience-driven leadership grounded in peer-recognised expertise. This form of emergent leadership shapes conversational norms, guides clinical interpretations, and contributes to the coherence and functioning of the community.” (lines 354–376).

  • AI and LLMs — positioning as future work

“Finally, the discussion on the relevance of CoPs in the age of Artificial Intelligence (AI) and Large Language Models (LLMs) is timely and compelling. The concept of a "dual infrastructure technological and human" is a strong concluding point.  The paper argues that AI can "accelerate literature retrieval and data synthesis" and that the CoP provides the critical interpretation. This is a theoretical assertion but lacks empirical backing from the current study. The survey did not assess the existing or planned use of AI tools. Since the study did not empirically explore the use or perceptions of AI/LLMs within the community, the discussion of the "dual infrastructure" should be positioned more clearly as a future strategic recommendation for DonnaRosa, rather than a conclusion drawn from the current data. Could the authors propose specific ways they plan to study the integration of AI tools within the CoP in the next phase of their research?”

We appreciate this important clarification. We have revised the AI-related section to emphasise that the discussion of a “dual infrastructure” (human and technological) is intended as a forward-looking reflection rather than an empirical conclusion from this dataset.

The revised text now clearly frames AI integration as a future strategic direction for the CoP and proposes concrete next steps, including:

  • surveying the community about AI/LLM usage and perceived utility;
  • piloting AI-assisted literature summarisation within DonnaRosa;
  • examining how AI tools could complement or challenge collective reasoning processes.

These revisions are included in the Discussion (lines 417–423).

Round 2

Reviewer 1 Report

Comments and Suggestions for Authors

the manuscript has been revised to contextualise the use of a CoP with other studies and the discussion is significantly improved highlighting the strengths and vulnerabilities of this tool which has organically grown in our profession without guidelines. the paper is greatly improved but i feel the abstract and simple summary should reflect the concerns regarding governance and responsibility and the need for society based guidelines for this area

Author Response

We thank the reviewer for highlighting this important omission in both the summary and the abstract. We have incorporated a brief statement addressing the need for society-level governance to ensure the safe and responsible use of instant-messaging tools in oncology.

Reviewer 2 Report

Comments and Suggestions for Authors

The authors have taken into account all the reviewers' considerations. I recommend publication of the manuscript.

Author Response

We sincerely thank the reviewer for the supportive comments and for recommending the manuscript for publication